

# Estimating the soil N$_2$O emission intensity of croplands in northwest Europe

Vasileios Myrgiotis[1], Mathew Williams[1], Robert M. Rees[2], and Cairistiona F. E. Topp[2]

[1]School of GeoSciences, University of Edinburgh, Edinburgh EH9 3JN, UK
[2]SRUC, West Mains Road, Edinburgh EH9 3JG, UK

**Correspondence:** Vasileios Myrgiotis (v.myrgiotis@ed.ac.uk)

**Abstract.** The application of nitrogenous fertilisers to agricultural soils is a major source of anthropogenic N$_2$O emissions. Reducing the nitrogen (N) footprint of agriculture is a global challenge that depends on our ability to quantify the N$_2$O emission intensity of the world's most widespread and productive agricultural systems. In this context, biogeochemistry (BGC) models are widely used to estimate soil N$_2$O emissions in agroecosystems. The choice of spatial scale is crucial because larger scale
studies are limited by low input data precision while smaller scale studies lack wider relevance. The robustness of large-scale model predictions depends on preliminary and data-demanding model calibration/validation while relevant studies often omit the performance of output uncertainty analysis and underreport model outputs that would allow a critical assessment of results. This study takes a novel approach on these aspects. The study focuses on arable Eastern Scotland; a data-rich region typical of Northwest Europe in terms of edaphoclimatic conditions, cropping patterns and productivity levels. We used a calibrated and
locally-validated BGC model to simulate direct soil N$_2$O emissions along with NO$_3$ leaching and crop N uptake in fields of barley, wheat and oilseed rape. We found that 0.59% ($\pm$0.36) of the applied N is emitted as N$_2$O while 37% ($\pm$6) is taken up by crops and 14% ($\pm$7) is leached as NO$_3$. We show that crop type is a key determinant of N$_2$O emission factors (EF) with cereals having a low (mean EF<0.6%), and oilseed rape a high (mean EF=2.48%), N$_2$O emission intensity. Fertiliser addition was the most important N$_2$O emissions driver suggesting that appropriate actions can reduce crop N$_2$O intensity. Finally, we
estimated a 74% relative uncertainty around N$_2$O predictions attributable to soil data variability. However, we argue that higher resolution soil data alone might not suffice to reduce this uncertainty.

## 1 Introduction

The emission of N$_2$O from agricultural soils is a particularly interesting aspect of the global N cycle because it represents a key contribution of modern agriculture to climate change which, in turn, poses a serious threat to agriculture itself (Paustian
et al., 2016). Nitrous oxide is a powerful greenhouse gas with a global warming potential 298 times stronger than carbon dioxide (CO$_2$) at a 100 years time horizon and it is also a major stratospheric ozone depleting substance (Galloway et al., 2013; Ravishankara et al., 2009). Almost a third of the N$_2$O entering the atmosphere on an annual basis originates from human activities with agriculture being the principal source (Reay et al., 2012). Nitrous oxide is emitted from agricultural soils as a direct consequence of the application of nitrogenous fertilisers, crop residues and manures but it can also be released indirectly



from leached N compounds (i.e. nitrate) (de Vries et al., 2011). The emission of $N_2O$ from agricultural soils is controlled by the microbe-mediated processes of nitrification and denitrification. These processes are tightly coupled and are affected by the combination of environmental conditions, the soil's physical and biochemical composition and the amount, timing and type of the applied fertiliser (Smith, 2017). These facts suggest that soil $N_2O$ emissions are highly variable both spatially and

temporally, which makes measuring and predicting soil $N_2O$ particularly difficult (Cowan et al., 2014). In its latest report, the Intergovernmental Panel on Climate Change (IPCC) stated that, on average, 1% of the N applied to agricultural soils is lost directly to the atmosphere as $N_2O$. Nevertheless, this default $N_2O$ emission factor (EF) is generic and does not reflect the role of different N amendments (e.g. ammonium nitrate, urea), soil conditions (e.g. texture, pH), climate (e.g. wet, dry) and cultivated crops, which vary greatly on a global scale but act together to define the real $N_2O$ EF (Leip et al., 2011).

Even though crop production occurs in almost every corner of the world certain areas are major crop producers (Monfreda et al., 2008). These areas are critical components of the global agricultural system and important consumers of nitrogenous fertilisers hence, they are also possible hotspots of soil $N_2O$ emissions (Potter et al., 2010). One of these areas is Northwestern Europe, which lies between $47^oN$ and $57^oN$ and includes Eastern Great Britain, Northern France, the Low Countries, Western Germany and the temperate plains of Scandinavia (Olesen et al., 2011). The main crops cultivated in this wide zone are cereals;

primarily wheat and secondarily barley (European Commission, 2018). Crop farming is intensive and depends on the use of synthetic fertilisers and other agro-chemicals to achieve wheat and barley yields that are the highest in the world (Stoate et al., 2001). The climate of the plains of Northwest Europe is oceanic and brown soils (Cambisols) are the most widespread soil group (Bonsall et al., 2002; Toth et al., 2008).

Scotland is a small yet representative region of Northwestern Europe. Almost all of Scotland's arable soils are located at

the eastern part of the country where climatic and soil conditions are favourable to the cultivation of crops (Hay et al., 2000). Crop production is dominated by wheat, barley, oilseed rape, oats and potatoes with wheat and barley alone representing three quarters of the total arable area (The Scottish Government, 2013). Most existing estimates of agricultural soil $N_2O$ emissions in this area are based on the default (Tier1) IPCC EFs and statistical models which do not consider many of the underlying $N_2O$-controlling factors (Lilly et al., 2003; Flynn et al., 2005; Lilly et al., 2009). The few model-based estimates on $N_2O$

emissions from Scottish croplands are part of national scale (UK) studies that consider all types of agricultural land use and do not examine the role of different crop types (Brown et al., 2002; Cardenas et al., 2013). Since 2016, the National Greenhouse Gas Inventory of the UK uses a country-specific (Tier 2) $N_2O$ EF equal to 0.791% for arable soils on which mineral fertiliser is added. Nevertheless, the estimates of agricultural soil $N_2O$ emissions remain the most uncertain aspect of greenhouse gas budget of Scotland and the UK (Bell et al., 2014).

Process-based agroecosystem biogeochemistry (BGC) modelling is widely used to estimate soil $N_2O$ emissions as well as other variables such as crop yields, $CO_2$, methane ($CH_4$) and $NO_3$ leaching. Such models, along with long-term country-tailored measurement inventories, are considered as the best tools for estimating $N_2O$ emissions from agricultural soils (i.e. IPCC Tier 3 method) (Sylvester-Bradley et al., 2015). Agroecosystem BGC models vary in complexity but the most advanced amongst them provide detailed descriptions of the processes that directly and indirectly affect the production on $N_2O$ in soils

(Necpálová et al., 2015; Gilhespy et al., 2014; Holzworth et al., 2014; Balkovia et al., 2013). Model-based estimates, and





especially estimates of soil $N_2O$, depend on the modelling scale and local conditions. The quality of input data can affect the robustness of model predictions, which for large scale simulations depends on the coarseness of the spatial input data (Pogson and Smith, 2015). Moreover, models are developed and parameterised using experimental data collected in the lab or at the field under conditions, which can be very different to those of the area in which the model is applied. Therefore, models have

to be tested at and calibrated to the climatic and soil conditions of their area of application (Li et al., 2015; Bellocchi et al., 2014; Wang and Chen, 2012). However, rarely do regional-scale model-based studies provide information on how (and if) the used model was calibrated and tested under the conditions (climatic, soil, crop types) of the region in which it was used.

Scotland is an interesting case from a BGC modelling perspective because (1) it is among the few European countries for which data of high spatial resolution (and quality) on climate, soil and crop management are available; (2) it maintains a na-

tional greenhouse gases inventory including measured data on soil $N_2O$ emissions from different types of croplands and (3) its croplands are made up of farms with a large average size (i.e. $>0.5km^2$) by Northwest European standards and, therefore, smaller crop cover uncertainty within each simulation spatial unit (i.e grid cell) (Eurostat, 2015). In this study, we build upon these advantages and apply a process-based agroecosystem model (Landscape-DNDC) across eastern Scotland to simulate soil $N_2O$ emissions from the arable soils (i.e. wheat, barley, oilseeds) of the region (Haas et al., 2012). The model's para-

metric sensitivity has been examined by Myrgiotis et al. (2018a) and it has been calibrated and evaluated under UK/Scottish edaphoclimatic conditions by Myrgiotis et al. (2018b).

The study takes a novel approach on aspects that are critical for the robustness of large-scale model-based predictions. Firstly, and in contrast to previous $N_2O$-focused studies, nitrate ($NO_3$) loss via leaching and crop N uptake are also simulated to provide a picture of the N budget (soil mineral/organic N, nitric oxide and $N_2$ fluxes not considered) of the agroecosystems

of arable eastern Scotland and to assure that $N_2O$ prediction is not made at the expense of other key variables (i.e. an unrealistic description of the N budget).Secondly, each crop and soil type is considered individually in terms of expected cropping patterns (i.e. sow/harvest dates) and fertiliser input (i.e. timing/frequency/intensity of fertiliser use) instead of merging crops and soils into aggregate groups (i.e. cereals/oilseeds and clayish/sandy), which is typical of large-scale modelling studies. Thirdly, we estimate the relative uncertainty around the simulated $N_2O$, $NO_3$ and N uptake as caused by variability in soil-related model

inputs and model parameters. Through this link between model estimates, data precision and model parameterisation we identify possible future research foci. The three variables considered in this study (crop N, $N_2O$, $NO_3$) represent a part of the N budget of a crop field. A complete N budget would require the inclusion of soil mineral N (SMN) and other N-based gas fluxes i.e. nitrogen oxides (NOx), nitrogen gas ($N_2$) and ammonia ($NH_3$) volatilisation. Much more N is lost from the soil-crop system as $N_2$, NOx and $NH_3$ than it is as $N_2O$, however, field measured data on non-$N_2O$ fluxes are scarce and

thus the scope for model evaluation is limited (De Vries et al., 2011). In regards to $NH_3$ volatilisation, most relevant data are collected at grasslands and soils treated with urea/manure while the present study focuses on arable soils (Baobao et al., 2016). Considering the aforementioned as well as the spatial and temporal scale of our study we have not considered these non-$N_2O$ gases. In summary, the aims of the study are to:

1. Estimate crop-specific $N_2O$ EFs in Scotland



2. Estimate the fraction of fertiliser N lost via NO$_3$ leaching and absorbed by crops

3. Quantify the parametric and input-data uncertainty around the model's predictions

## 2 Materials and methods

### 2.1 Regional simulations

5 A State-of-the-art process-based agroecosystem BGC model (Landscape-DNDC) was coupled to geographically explicit information on climate, soil and crop management in order to simulate soil N$_2$O emissions from the croplands of eastern Scotland (Haas et al., 2012). Leaching of NO$_3$, crop yields and crop N uptake were also simulated. The model was run using input data for 2011, 2012 and 2013; with 2013 being the year of reference for the results. The simulations for years 2011 and 2012 were performed to assure the stabilisation of the model's soil carbon (C) pools as suggested by the model's developers and proven 10 by preparatory simulations.

This study considered four crops that in 2013 represented 83% of the total arable area of Scotland and 93% of the arable area of eastern Scotland: winter wheat, winter barley, spring barley and oilseed rape. Only the application of ammonium nitrate was considered since this is the main source used in crop farming systems in Eastern Scotland. The spatial scale of the input data and of the outputs is 1 km$^2$ and the simulated area covers 3800 km$^2$. The model was run twice at regional scale: (1) using zero 15 fertiliser application (i.e. control treatment) and (2) using an amount of fertiliser that varies according to crop and soil type. For each grid cell, the difference between the total simulated emission of N$_2$O (kg N-N$_2$Oha$^{-1}$), NO$_3$ leached (kg N-N$_2$Oha$^{-1}$) and crop N uptake (kg N ha$^{-1}$) under control and treatment condictions was divided by the total amount of N (kg N ha$^{-1}$) applied to that grid cell and the results were expressed as factors:

- N$_2$O emission factor (N$_2$O EF)

20 - NO$_3$ leaching factor (NO$_3$ LF)

- Crop N uptake factor (NUF)

### 2.2 Landscape-DNDC

Landscape-DNDC is a complex process-based ecosystem model that describes the biogeochemistry of terrestrial ecosystems (Haas et al., 2012). It belongs to a large family of models who trace their conceptual background to the original DeNitrification-25 DeCompositon model (DNDC) (Gilhespy et al., 2014; Li et al., 1992). The model can simulate energy, water and nutrient transport inside the soil-plant-atmosphere system in arable, grassland and forest ecosystems. It has a modular structure that facilitates the integration of modules that describe different parts of the simulated system (i.e. plant growth, water cycling, soil biogeochemistry). Landscape-DNDC was developed with regional scale application in mind and facilitates the performance of multiple point-based (i.e. an area of 1ha) simulations in parallel.




The user has access to the model's parameters and can also reshape its structure by using different module combinations. The model requires input data on climatic (e.g. min/max precipitation, temperature, wind speed) and soil (e.g pH, clay content, bulk density) conditions as well as information on field management (e.g. crop rotation, date/depth of tillage). Soil discretisation is used to provide the model with soil-related information allowing all properties (e.g. soil BD, pH etc) to be attributed to

layers of user-defined thickness (cm). The model's crop growth module is simplistic (empirical/statistical) and depends on few parameters. For the purpose of this study the crop-growth module was calibrated using measured crop yield and N content data for barley, wheat, oilseed rape. These data were collected at UK-based experiments and extracted from the relevant literature.

Landscape-DNDC was run using the default values for all of its soil biogeochemistry and hydrology-related parameters (i.e. more than 100 unique parameters) except for nine parameters. These parameters (Table 1) were found to be critical for

the prediction of $N_2O$ fluxes and other aspects of N cycling (including $NO_3$ and crop N uptake) under UK edaphoclimatic conditions (Myrgiotis et al., 2018a, b). Table 1 presents the nine soil biogeochemistry parameters, their values and respective ranges that were used for the simulations.

**Table 1.** Most critical soil biogeochemistry parameters of the Landscape-DNDC model.

| Parameter | Name | Range | Value used |
|---|---|---|---|
| Microbial death rate | AMAXX | 0.71 - 1.95 | 1.72 |
| Microbial denitrifier fraction | DENIFRAC | 0.46 - 0.88 | 0.73 |
| Fraction of decomposed carbon that goes to the dissolved organic carbon pool | EFFAC | 0.40 - 0.93 | 0.71 |
| Reduction constant for $N_2O$ diffusion | EFF $NO_2$ | 0.22 - 0.59 | 0.34 |
| Reduction constant for NO diffusion | D NO | 0.046 - 0.105 | 0.0709 |
| Microbial efficiency for $NO_2$ denitrification | D $N_2O$ | 0.038 - 0.091 | 0.073 |
| Reaction rate for nitrification | KNIT | 0.6 - 9.0 | 2.25 |
| Microbial growth rate | MUEMAX | 2.470 - 6.000 | 3.07 |
| Microbial growth rate for denitrification on $NO_2$ | MUE $NO_2$ | 0.395 - 1.000 | 0.79 |

## 2.3 Uncertainty quantification

Model-based predictions are subject to uncertainties caused by input data and the model's parameters and structure (Norton,

2015). The effect of uncertainty around soil-related model inputs and model parameters on the prediction of $N_2O$, $NO_3$ and crop N uptake was quantified in this study by running the model in 1500 grid cells, which were selected using random sampling weighted according to soil texture (i.e. pseudo-regional simulations). As a first step, the model was run at the pseudo-regional scale 300 times using at each iteration (for each soil cell) a vector of input values (for bulk density (BD), clay fraction, pH, C content, wilting point, field capacity) that was created by randomly sampling the respective variable's range (i.e. standard

deviation of measurements). In a second step, the model was run again 300 times using at each iteration the previously-used



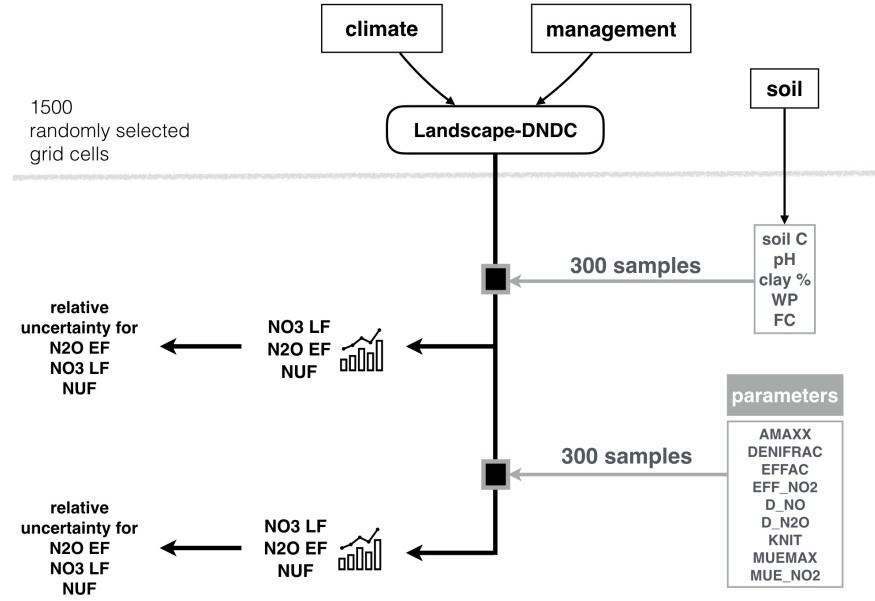

**Figure 1.** Schematic description of the process that is used to quantify the relative uncertainty of regional predictions. EF: emission factor, LF: leaching factor, NUF: N uptake factor, WP: soil wilting point, FC: soil field capacity. For descriptions of the 9 parameters see table 1.

soil input vectors and 300 additional randomly sampled vectors for the nine soil biogeochemistry parameters (Table 1). Figure 1 presents the process that was followed to quantify soil input and parameter-related uncertainty over the model's predictions. The use of a condensed version of the full set of spatially distributed drivers (i.e. pseudo-regional setup) was devised as a low computer-demand method to assess prediction uncertainty at regional scale. The relative uncertainty was estimated for each grid cell by dividing the standard deviation for $N_2O$ EF, $NO_3$ LF and NUF by the respective mean. The relative uncertainty is expressed in the results section as a percentage.

## 2.4 Regional inputs

The information that was used to drive the model at regional scale consisted of data on climate, soil, crop cover and management (see Table 2). Only the crop cover data were spatially processed in order to produce information at a 1 km$^2$ grid, which is the spatial scale of reference for the simulations and the results of this study.

### 2.4.1 Climate data

The climate data were obtained from the Climate, Hydrological and Ecological research Support System (CHESS) of the Centre for Ecology & Hydrology of the UK (Robinson et al., 2017). The data used in this study include daily precipitation (mm) and minimum and maximum temperature ($^o$C). The original spatial resolution of the data was 1 km$^2$.





**Table 2.** Regional scale input data sources

| Type | Source | Variables | Temporal Resolution | Spatial Resolution |
|---|---|---|---|---|
| Soil | Scottish Soil Database | soil depth, pH, C, bulk density, clay content, wilting point and field capacity | N/A | 1 km$^2$ |
| Crop | EDINA AgCensus | harvested area per crop type | 2011-2013 | 4 km$^2$ |
| Climate | CHESS | precipitation, max/min T | 2011-2013 | 1 km$^2$ |
| Management | HGCA crop growth guides Fertiliser Manual (RB209) | planting/harvest dates, recommended fertiliser rates, fertiliser application dates | N/A | N/A |

### 2.4.2 Soil data

A number of global-scale soil data products are available at different spatial resolutions (Batjes, 2016; Hengl et al., 2014). Scotland maintains its own soil database, which was constructed based on field surveys. Therefore, the soil data used in this study were extracted from the Scottish Soil Database (Scottish Spatial Data Infrastructure, 2014). The data present the

distribution of different soil classes across Scotland in a gridded format (1 km$^2$). More than one soil class can be found within the limits of a single grid cell. For each soil class, information is provided about the soil's measured depth (m), the number of soil layers and the thickness (m) of each layer. For each layer the mean and standard deviation of the measured bulk density (BD), clay content (%), soil C content (%), soil pH, wilting point and field capacity are provided along with the type of the soil's texture. It should be noted here that data of the Scottish Soil Database can be obtained at a very fine (250m) resolution.

Our decision to use the 1 km$^2$ resolution data was based on the fact that (1) more than half of all crop farms in Scotland are larger than 0.5 km$^2$ in size and close to one third are larger than 1 km$^2$, and that (2) the granularity of the 250 m resolution data is not lost when the data are aggregated to the 1 km$^2$ grid (i.e. number/coverage of different soil groups).

### 2.4.3 Crop cover data

The crop cover data were obtained from the EDINA AgCensus database (Edina AgCensus, 2016). The data define the area

(in km$^2$) that winter wheat, spring/winter barley and winter oilseed rape covered in 2011, 2012 and 2013. The original spatial resolution of the crop cover data was 4km$^2$ (Fig 2). The data were spatially disaggregated and linked to the soil and climate data (Fig 3).




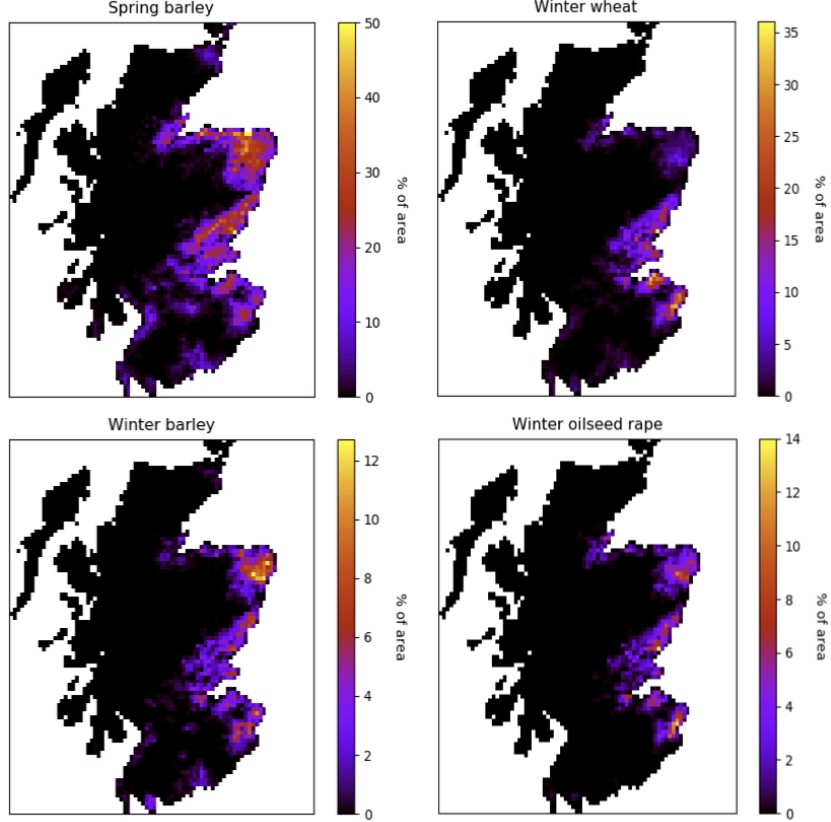

**Figure 2.** Percentage of area covered by each crop type in Scotland (4 km$^2$ grid).

### 2.4.4 Crop management data

The crop management data were compiled from information in the DEFRA fertiliser manual (DEFRA, 2010) and the crop growth guides published by the UK Home-Grown Cereals Authority (HGCA). The information that was obtained from these sources referred to the four crops considered in the study (winter/spring barley, winter wheat, winter oilseed rape) and included:

  – Planting dates for each crop type

  – Harvest dates for each crop type

  – Total recommended amount of fertiliser applied to each crop type depending on soil texture and depth

  – Number of separate treatments (splits) in which the total amount of fertiliser was applied to the soil (per crop type)

  – Date of application of each fertiliser split (per crop type)

  – Percentage of total recommended fertiliser that was applied at each split application (per crop type)





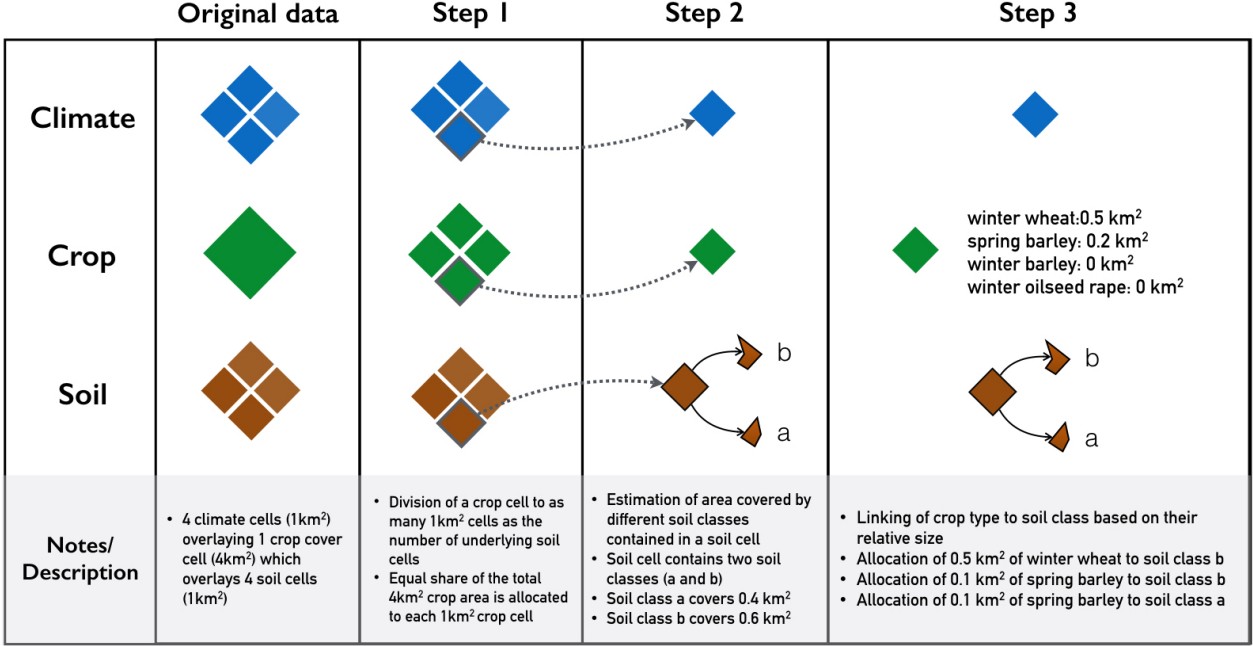

**Figure 3.** Description of the processing of spatial data on climate, soil and crop cover. Example values are used for the size (km$^2$) of soil classes (step 2) and the area covered by different crops (step 3) to demonstrate the process followed.

Further details on the simulated management activities (e.g. harvest dates, fertiliser kgNha$^{-1}$ etc) are presented in the appendix.

# 3 Results

## 3.1 Crop yields

5 We divided the total simulated yield to the total simulated area (per crop type for 2013) and compared the results with respective data from the Scottish agricultural census of 2013 (Fig 4). While the model consistently overestimates yields per hectare for cereals, the simulated yields appear to be in the right order of magnitude for all crops. The simulated cereal yields are on average 10% higher than the observed value with the difference ranging between 20% for winter barley and 8% for winter wheat.

## 3.2 N$_2$O

10 The simulated mean N$_2$O EF for arable eastern Scotland was 0.59% ($\pm$0.36%) with 75% of the EFs estimated being below 0.70% (Fig 5). The results of the simulations are presented as EFs and thus integrate the amount of fertiliser N (kgNha$^{-1}$)





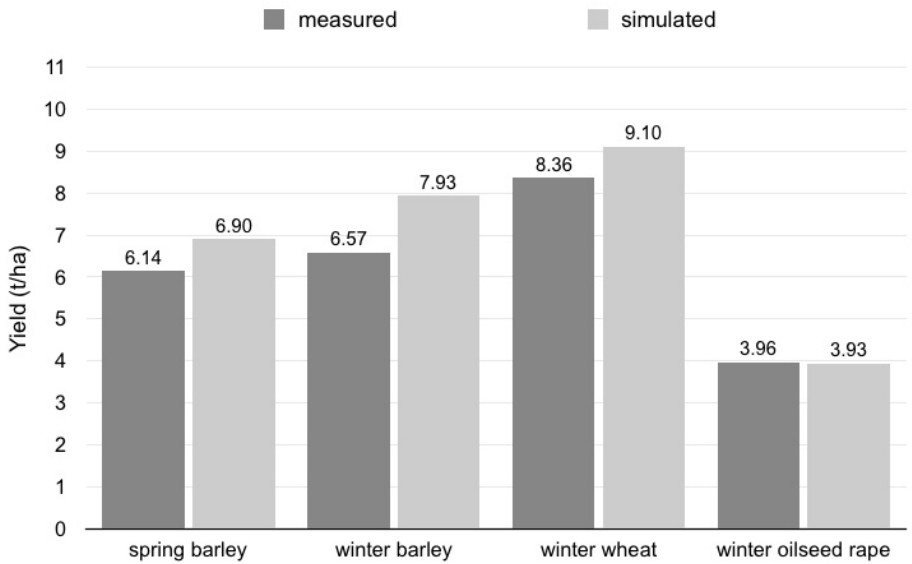

**Figure 4.** Comparison of mean measured yields (tn ha$^{-1}$) for Scotland and simulated yields (tn ha$^{-1}$) values for Eastern Scotland.

applied at the soils present in each 1 km$^2$ cell. In this context, the spatial distribution of the esimated N$_2$O EFs (Fig 6) reveals one sizeable hotspot of high EFs (area in red at northeast part of the map).

A closer analysis of the N$_2$O EFs revealed a stratification of N$_2$O EFs that is related to crop type. By grouping N$_2$O EFs according to crop (Table 3) the dominant presence of spring barley in the simulated area becomes apparent; with the mean estimated N$_2$O EF for all crops (i.e. 0.59%) being very close to the EF that is specific to spring barley (0.57%). Winter wheat and winter barley, which cover almost a third of the simulated area, were found to have a lower crop-specific EF than spring barley. The N$_2$O footprint of winter oilseed rape (WOSR) was very high (2.48%) and, in most cases, was responsible for the high N$_2$O hotspots (Fig 6).

Considering the fact that soils of loamy and sandy loam texture represent most of the simulated soils, the grouping of the estimated EFs according to soil texture (Fig 4) showed that N$_2$O EFs were slightly higher in loamy soils than in sandy loam. Even in light of this information the average N$_2$O EF was $\approx 0.5\%$. Soil textures associated with low hydraulic conductivity like clay loams and silty clay loams produced higher-than-average EFs even though clay soils did not produce similar results. However, soils with clay texture represented only 0.1% of the total simulated area.

### 3.3  NO$_3$ and crop N uptake

The simulated mean NO$_3$ LF for arable eastern Scotland was 14% ($\pm$7%) with 75% of the estimated NO$_3$ LFs being less than 18% (Fig 7). In terms of the simulated uptake of N by crops, the mean NUF was 37% ($\pm$7%) with three quarters of the estimated NUF being below 40% (Fig 8).



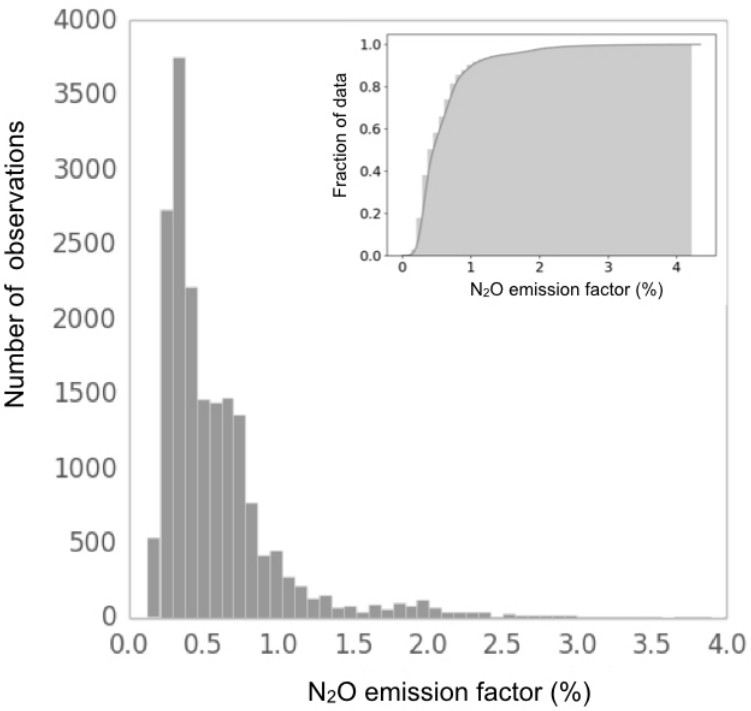

**Figure 5.** The distribution and cumulative distribution (small graph) of simulated $N_2O$ EFs

**Table 3.** Estimated $N_2O$ EF by type of crop.

| Crop Type | Share of total simulated area (%) | $N_2O$ EF (%) Mean | SD |
|---|---|---|---|
| winter wheat | 20 | 0.25 | 0.11 |
| winter barley | 10 | 0.34 | 0.18 |
| spring barley | 62 | 0.57 | 0.58 |
| winter oilseed rape | 8 | 2.48 | 0.85 |

## 3.4 Correlation of model drivers and outputs

The simulations that were performed to obtain the outputs which were then processed and expressed as emission, leaching and uptake factors produced a large volume of data that were further analysed to construct a heatmap (Fig 9) of the correlations between model drivers (e.g. soil pH, C etc) and outputs (e.g. yield, $N_2O$ etc). The soil-related drivers that were used to produce

5    the heatmap refer to the top 30 cm. A strong (>0.7) positive correlation was observed between the predicted crop yield and N





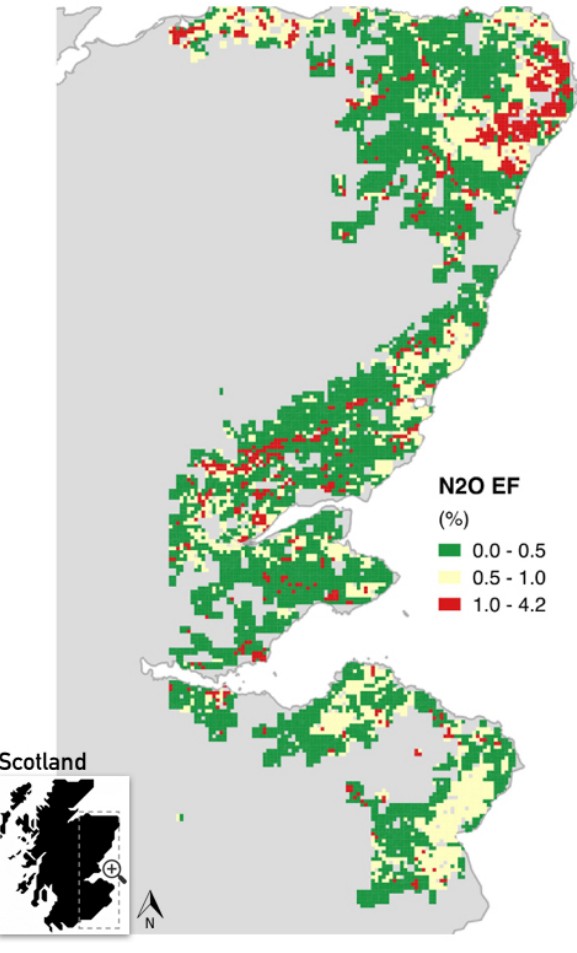

**Figure 6.** Geographic distribution of simulated N$_2$O EFs in Eastern Scotland.

uptake. This can be explained by the coupling of crop N absorption and biomass growth both in reality and according to the concept embodied in Landscape-DNDC. Total NO$_3$ leaching (but not the associated NO$_3$ LF) shows a positive correlation with crop yield and uptake as they do with the rate of fertiliser application. This is explained by the fact that the loss of N from the soil via leaching is driven by the amount of N present in the soil and thus depends strongly on the fertiliser rate.

5      The estimated N$_2$O EFs were negatively correlated to NUF and NO$_3$ LF which is explained by that fact that N loss via leaching and crop uptake act in opposition to the emission of N$_2$O (and other N-based gases). Also, high soil field capacity and high C content appear to have a positive impact of the esimated N$_2$O EF and N$_2$O emission in general. As the capacity of a simulated soil area to hold water increases (i.e. high FC low WP) so does the possibility of larger anaerobic soil zones and therefore of denitrification actitivity. The size of a soil's microbial pool depends (in modelling terms) on its C content

10    and therefore more C results in larger microbial population. This in turn leads to an increase in the presence of nitrifying and denitrifying organisms and the production of N$_2$O that they control. This process complexity -where factors interact to generate





**Table 4.** Estimated $N_2O$ EF by type of soil texture. Soil textures are listed in order of decreasing hydraulic conductivity.

| Soil Texture | Share of total simulated area (%) | $N_2O$ EF (%) Mean | SD |
|---|---|---|---|
| Sandy | 1.1 | 0.44 | 0.28 |
| Loamy Sand | 0.9 | 0.43 | 0.19 |
| Sandy Loam | 50.0 | 0.54 | 0.66 |
| Sandy Clay Loam | 2.3 | 0.41 | 0.71 |
| Loam | 39.0 | 0.73 | 0.9 |
| Clay Loam | 4.8 | 0.93 | 1.02 |
| Silty Clay Loam | 0.9 | 1.02 | 1.07 |
| Clay | 0.1 | 0.64 | 0.31 |

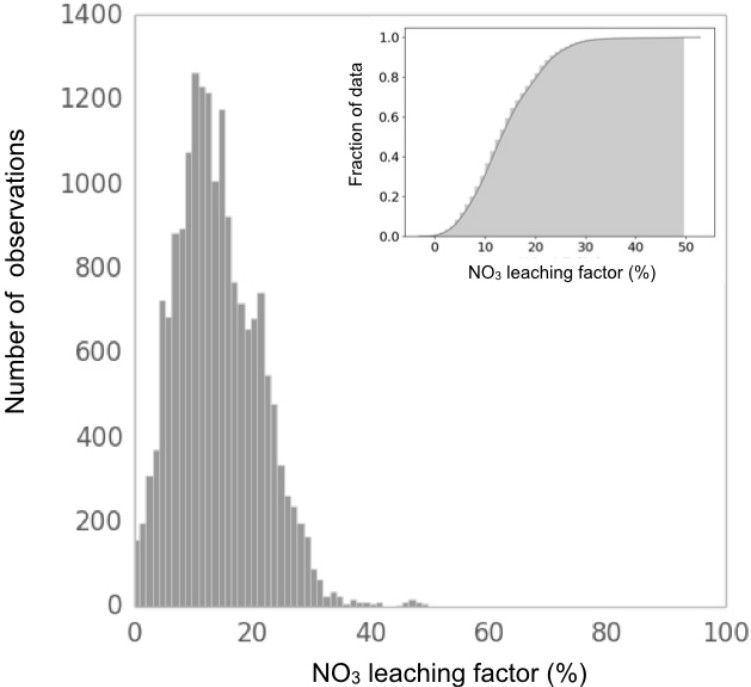

**Figure 7.** The distribution and cumulative distribution (small graph) of simulated $NO_3$ LFs

spatial and temporal variability- highlights the value of process-based modelling in predicting soil $N_2O$ emissions. Overall, the fertiliser rate stands out as the only input variable that clearly affects all three N-based outputs as well as yield.





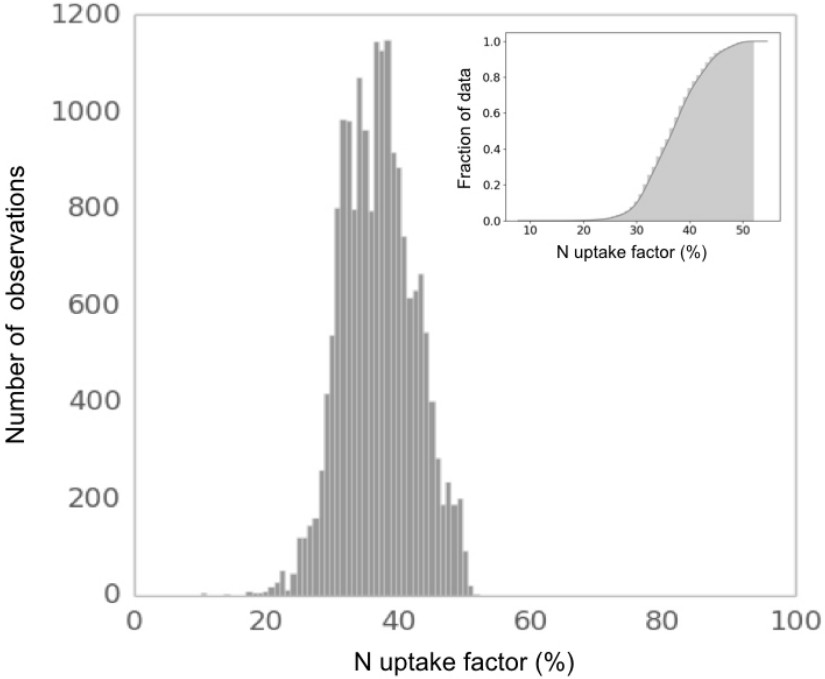

**Figure 8.** The distribution and cumulative distribution (small graph) of simulated NUFs

### 3.5 Relative uncertainty of model predictions

We used 300 randomly sampled vectors of soil-related inputs and model parameters to run the model at a pseudo-regional scale in order to provide an assessment of the input and parametric uncertainty of the estimated $N_2O$ EF, $NO_3$ LF and NUF (Fig 1). The results of this process are summarised in the bar plots presented in Figure 10. The relative uncertainty values express the

5 mean relative standard deviation that was estimated at 1500 randomly selected grid cells after (1) 300 simulations performed under varying soil inputs (i.e. clay, pH, C, WP, FC) only and (2) 300 simulations performed under varying soil inputs and model parameters. The stabilisation of the simulated outputs relative to the size of samples used has been assessed visually by plotting the pseudo-regional mean and SD for each variable ($N_2O$, $NO_3$ and NUF) against the sample size.

The results (Fig 10) show that the $N_2O$ EF is the output variable most sensitive to uncertainties around model inputs in

contrast to NUF which appears to be insensitive to them. The relative uncertaintly of the simulated $NO_3$ LF is 25% smaller than that of $N_2O$ EF. The impact of uncertainty of the nine key model parameters (related to soil biogeochemistry) appears to be small compared to that of soil-related inputs. The small number of parameters examined in this uncertainty analysis and the fact that their ranges have been calibrated/constrained according to UK conditions (i.e. measured $N_2O$ data) can explain their limited role. However, these 9 biogeochemical parameters have a noticeable impact on $NO_3$ leaching prediction. We attribute

this higher parametric sensitivity to the fact the parameters have been calibrated only against measured soil $N_2O$ data. We can reasonably assume that the use of (currently lacking) measured data on $NO_3$ leaching would have had a similar impact on $NO_3$





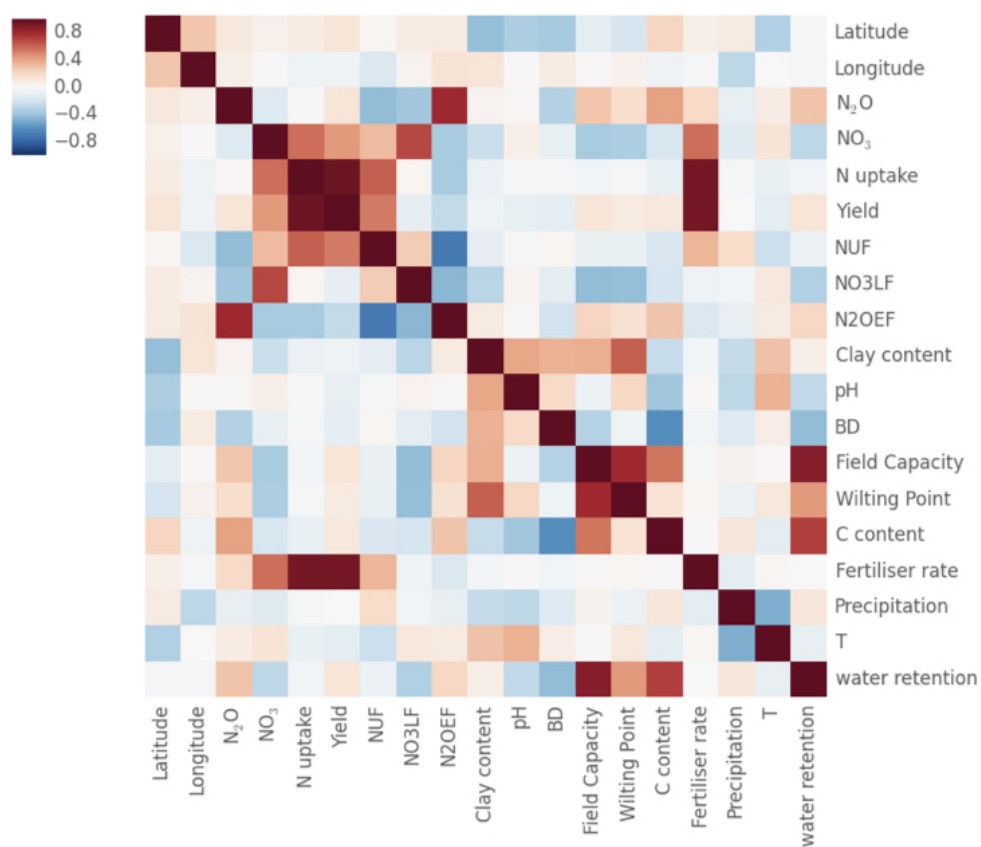

**Figure 9.** Heatmap of correlations between model inputs and outputs and estimated factors.

LF relative uncertainty. Overall, our analysis was not a full quantification of the model's parametric uncertainty but a targeted analysis where only already-indentified parameters were included (Myrgiotis et al., 2018a). The results suggest that model predictions of $N_2O$ emission and $NO_3$ leaching are sensitive to uncertainties around soil-related input data such as soil clay content, pH, C and water retention (i.e. equal to FC minus WP) properties with $N_2O$ emissions being particularly sensitive to
5   them.

## 4   Discussion

The mean $N_2O$ EF for Scottish croplands of 0.59% that was estimated in this study is smaller than the generic IPCC $N_2O$ EF (1%). However, this result is consistent with recent measured data in the UK (Bell et al., 2015; Sylvester-Bradley et al., 2015). A few studies have estimated $N_2O$ EFs from Scotland's croplands highlighting how these vary from year to year and how they
10   differ from default Tier 1 EFs (Dobbie et al., 1999; Hinton et al., 2015). Brown et al. (2002) used process-based modelling on a county-level spatial scale to derive a cropland-specific EF of 1%. Flynn et al. (2005) analysed and upscaled field-scale $N_2O$



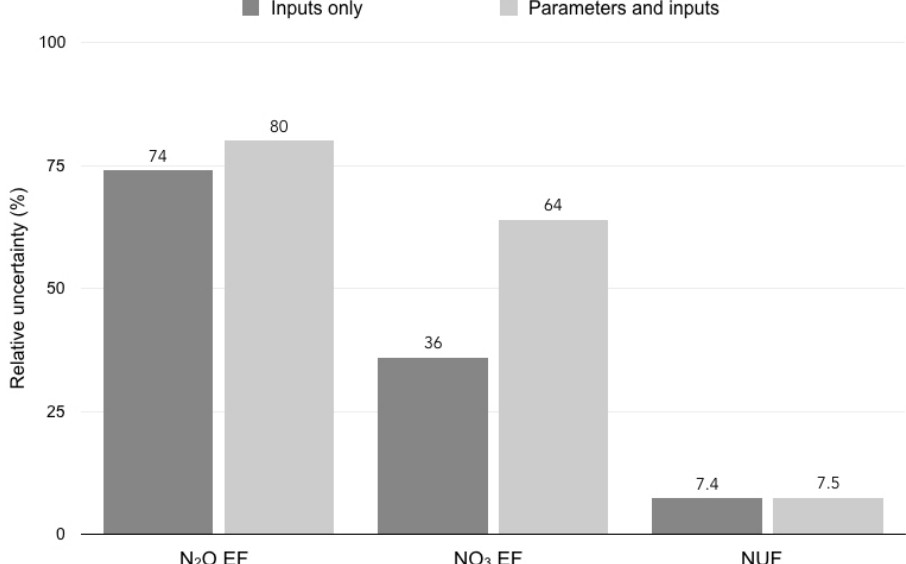

**Figure 10.** Uncertainty (expressed as relative uncertainty) of predicted $N_2O$ EF, $NO_3$ LF and NUF as caused by the (1) variability in soil-related input data (inputs only) and (2) the variability in soil-related input data and model parameters (parameters and inputs).

measurements to estimate an arable-specific $N_2O$ EF equal to 0.6% resulting from mineral fertiliser application. Cardenas et al. (2013) run the UK-DNDC model using gridded input data (5 km resolution) and found an $N_2O$ EF of 0.82% (±0.48) under synthetic fertiliser application and 0.44 (±0.33)% for manure application. Lilly et al. (2009) used modelled soil wetness data (at 100 m resolution) to upscale measured soil $N_2O$ data. They estimated an average cereal-specific $N_2O$ EF of 0.79 % (0.66-

1.08 % for spring cereals and 0.50-0.75 % for winter cereals). More recently, Fitton et al. (2017) simulated $N_2O$ emissions in UK croplands using the DailyDayCent model and estimated the contribution of different factors (i.e. soil properties, climate, fertiliser rate) to the simulated $N_2O$. In common with our study (Fitton et al., 2017) found that fertiliser rate was the most important factor contributing by 60% to the total simulated $N_2O$ emissions in Scotland (Fig 9).

   A number of studies have presented crop-specific soil $N_2O$ EFs in other parts of Northwest Europe. Gabrielle et al. (2006)

used process-based modelling (i.e. Crop Environmental REsources Synthesis model) to simulate soil $N_2O$ emissions in the most important wheat-producing region of Northern France. The authors highlighted the role of the spatial resolution of available input data in model-based studies and estimated a winter wheat-specific $N_2O$ EF that was less than a third (i.e. 0.07-0.30%) of the default IPCC EF; for comparison, the winter wheat-specific EF in our study was 0.25% (±0.11). Dechow and Freibauer (2011) used empirical modelling based on a measured dataset (from Northwest European sites) to estimate a mean cropland-

specific $N_2O$ EF of 0.91% for Germany. The measured data that Dechow and Freibauer (2011) used was a subset of the dataset compiled by Stehfest and Bouwman (2006) whose study also suggested a global cropland-specific $N_2O$ EF of 0.91%. The Dechow and Freibauer (2011) study also found that the $N_2O$ EF followed a southeast to northwest gradient (i.e. from continental to oceanic climate) with EFs in North and West Germany being between 0.25 and 0.75%; significantly lower than





those in the South and East. Lesschen et al. (2011) also used the Stehfest and Bouwman (2006) dataset to estimate $N_2O$ EFs tailored to European conditions and stratified according to land use, texture and fertiliser type. Their study suggested that between 0.4 and 0.6% of the N (in Ammonium Nitrate form) applied to arable soils is lost directly as $N_2O$ (excluding emissions due to crop residues N). The lower EF (0.4%) corresponds to more sand-based soils and the higher EF (0.6%) to

more clay-based soils. In one of the few studies that present spatially-stratified soil $N_2O$ EFs for European croplands (i.e. 31 crops considered together), Leip et al. (2011) used European-wide input data and a process-based model (DNDC-EUROPE) to estimate mineral fertiliser-specific $N_2O$ EFs equal to 0.5% for the UK, 0.9% for France and 1.7% for Germany.

The results of our simulations showed that highest $N_2O$ EFs were associated with winter oilseed rape with intermediate EFs occuring under spring barley and lowest $N_2O$ EFs under winter wheat and winter barley. The large $N_2O$ footprint of WOSR

production is caused by three factors. Firstly, the N content of the straw of WOSR is higher than that of the other crops (Walter et al., 2014). This means that more N is contained in the residues that are left on the field after harvest and thus more N is free to be emitted as $N_2O$ flux (Walter et al., 2014; Rathke et al., 2006). Secondly, oilseed rape requires more fertiliser N than cereals, therefore, more N is added to soils where WOSR is cultivated (Sylvester-Bradley and Kindred, 2009). Thirdly, crop growth representation in Landscape-DNDC is simplistic (semi-empirical) and treats oilseed rape more like a cereal. Oilseed rape

requires more N than cereals do at their early post-winter growth stage. For this reason the recommended amount of fertiliser N for WOSR at the first (out of two) splits is significantly higher than that of cereals (Bouchet et al., 2016). Consequently, the semi-empirical crop growth module tends to underestimate crop N uptake by WOSR right after the first application (i.e. March) leaving more N to be lost as $N_2O$ and $NO_3$. A high $N_2O$ footprint for oilseed cultivation in Northwest Europe has been identified in studies in France (Jeuffroy et al. 2013) and Germany (Kaiser and Ruser 2000) as well as in a meta-analysis

by Walter et al. (2014) and a model-based study by Flynn et al. (2005). The high $N_2O$ emissions associated with oilseed rape cultivation in the UK have also been discussed by (Baggs et al., 2000).

In order to provide a more complete picture of N budgeting in the simulated agroecosystems we estimated the fraction of applied N that is leached as $NO_3$ LF and absorbed by the growing crop (NUF). In this context, the predicted mean $NO_3$ LF (14%) was almost half the generic LF suggested by IPCC (30%) and the UK cropland-specific LF suggested by Cardenas et al.

(2013) (28%). According to Landscape-DNDC, leached N is the amount of N (kg N ha$^{-1}$) that leaves the lowest layer of the simulated soil. A regional-scale study using Landscape-DNDC in Germany has found that total mean $NO_3$ leaching exceeds the total mean $N_2O$ emissions by a factor of 20 (Klatt et al., 2016). In our study this factor was equal to 24. Possible reasons for the discrepancy between modelled leaching losses and observations may include pathways of N loss that are not fully characterised by the model (e.g. surface flow) or a tendency of the model to overestimate soil N storage. In terms of crop N uptake, the mean

NUF (37%) is slightly above the global average N use efficiency (NUE) estimated by Raun and Johnson (1999) (i.e. 33%). It should also be clarified that by NUE we refer here to the value that is estimated according to Raun and Johnson (1999) where only fertiliser-based N is considered as a N source. The global NUE suggested by Raun and Johnson (1999) was produced using data from a large set of countries and integrates high and low efficiency crop-growing systems. A more recent study by Lassaletta et al. (2014) has found that the global NUE has been in decline from 1961 (65%) to 1980 (45%) and has since then

stabilized at 47%. Lassaletta et al. (2014) also estimated that NUE in the Netherlands, which is a Northwest European country,



was around 35%. In a UK context, measurements of fertiliser recovery by spring barley in Scotland estimated N recoveries between 43% and 61% (McTaggart and Smith, 1995). Also, Sylvester-Bradley and Kindred (2009) analysed data on N uptake for various UK crops and showed that around 60% of the total available N (i.e. N in soil and in fertiliser) is taken up by the crop; if only spring and winter barley, winter wheat, winter oilseed rape are considered. The respective value based on our regional simulations was 59%, which means that roughly 2/3 of crop N comes from feriliser applied during the growing season and 1/3 comes from the soil's N pool; itself shaped by post-harvest residues decomposition and wet/dry deposition. Overall, and taking into account the fact that the crop growth module of Landscape-DNDC is relatively simplistic, the model tended to under-predict NUF but, nevertheless, provided crop N contents within realistic bounds.

The estimated sensitivity of simulated soil $N_2O$ emissions to soil-related model inputs suggests that the results and conclusions of this study depend strongly on the precision of the spatial soil data that were used. This dependency is more powerful than that on the model's parameters. The parametric sensitivity of $NO_3$ leaching prediction is almost equal to its sensitivity to soil inputs but we consider this to be an artefact of model calibration against measured $N_2O$ data only. The low sensitivity of the predicted crop N uptake to model parameters and soil inputs reflects the fact that, under sufficient N suppply, climate is the main determinant of crop growth and, consequently, N uptake. In general, we believe that soil input-related uncertainty could be reduced by using higher resolution soil data ($<1km^2$). However, the uncertainty that is caused by management-related inputs (i.e. sow/harvest timing, fertiliser splits and amounts) should be expected to remain rather large even when using higher resolution data. This is because management-related inputs are constructed by combining crop cover and soil data layers whose original resolution is 4 and $1km^2$ respectively. As this study showed, fertiliser use is a key driver of simulated soil $N_2O$, $NO_3$ leaching, crop N uptake and yields. Taking everything into account, we argue that the use of high resolution soil data would be more effective when combined with accurate information on crop cover (i.e. satellite-derived crop cover information) because that would allow more accurate calculations of the timing and the amount of fertiliser application in the simulated crop fields.

The method used to quantify model input and parametric uncertainty in this study was designed so as to be time efficient but at a minimum cost in terms of robustness. We estimated that a full regional-scale uncertainty analysis would have lasted 27 times more than what our pseudo-regional analysis did. Furthermore, all simulated agroecosystems (i.e. soil-crop combinations) are managed according to recommended practices with activities like harvest and fertiliser application happening at the same time across large areas. This synchrony is not realistic and it is done purely due to lack of accurate knowledge of actual management. Management interventions that under certain climatic conditions can lead to increased loss of N can be represented in simulations even though in actuality they would be rescheduled (e.g. fertiliser spread on a day of very heavy rainfall). It can be argued that the simulated systems are managed less intelligently than real-life systems. Nevertheless, the large number of simulations (18367 unique instances) under a range of climate, soil and crop management combinations together with the fine spatial scale of the input data assure that any patterns in soil $N_2O$ production (e.g. localised hotspots, strong dependency on certain drivers etc) are detectable.



## 5  Conclusions

This study offers new insights on the regional $N_2O$ footprint of arable agriculture in Scotland and, by extension, in Northwest Europe. It shows (1) that the soil $N_2O$ EF for cereals is well below the generic IPCC EF of 1%; (2) that winter oilseed rape cultivation (especially if in monoculture) contributes to much higher $N_2O$ emissions (per unit area) than that of cereals and (3) that the soil $N_2O$ emissions are the result of dynamic interactions between different factors, of which only fertiliser N was identified as having a clear effect on all simulated variables. In general, we conclude that, while the N footprint of northwest European croplands is affected by inherent edaphoclimatic conditions, human decisions on cropping patterns and fertilisation largely control direct soil $N_2O$ emissions. Moreover, the low mean crop N uptake that was estimated here suggests that there is considerable margin for increases in the N use efficiency of crop-based agricultural systems. A complete impact assessment of different N-conscious agricultural practices (e.g. reduced fertiliser use, alternative crop coverage patterns) would require a consideration of both crop yields and N-related variables ($N_2O$, $NO_3$, N uptake); but this was beyond the scope of our stydy.

We carefully considered critical aspects of large-scale agroecosystem BGC modelling by ensuring (1) that the model that was used was calibrated and tested under local conditions; (2) that a high resolution was selected for the input data considering their different original resolutions (3) that $N_2O$ fluxes were not simulated at the expense of a realistic prediction of the other aspects of cropland N budget and (4) that a measure of model output uncertainty was estimated and presented. These modelling tasks might be data-demanding but, we argue that, their completion guarantees the robustness of model-based studies and should, therefore, be part of similar studies in the future. Moreover, our uncertainty analysis results show that parameter calibration can minimise their impact on model output uncertainty leaving soil data as the main source of output uncertainty (when not considering model structural uncertainty).

Process-based BGC models will always represent simplifications of real agroecosystems but they offer the opportunity to explore management and policy interventions that can increase the resource-use efficiency and reduce the environmental footprint of arable systems across the world. The development of algorithms that reproduce farmer decisions (planting, fertiliser application etc) based on a combination of observed and simulated variables (e.g precipitation, temperature, crop/soil N content), and field-based information (e.g. census questionnaires) can improve agroecosystem modelling by replacing more simplistic approaches in defining the modelled field management with approaches that assume management is reactive to varying local conditions and even data-based. We believe that the development of such tools would benefit large-scale BGC modelling and could have synergies in the precision agriculture domain. In this context, remote sensing is increasingly used to obtain information on the crop cover and even crop N content. By combining such data with higher-resolution ($<1km^2$) soil data, which are available for Scotland, the soil input-related uncertainty of $N_2O$ model predictions can be reduced.

*Competing interests.*  TThe authors declare that there is no conflict of interest regarding the publication of this article.





*Acknowledgements.* The authors are grateful to the Scottish Government for supporting this work and to the NERC Greenhouse Gas Emissions & Feedbacks program for its contribution.

## Appendix A

**Table A0.** Details of the dates (julian day) of management activities (harvest, tillage, planting) and fertiliser application used in the simulations.

| Soil Type | Location | Crop | Tillage (julian day) | Planting (julian day) | Fertiliser (kgNha-1) | Splits | Fertiliser Date 1 (julian day) | Fertiliser Amount 1 (kgNha-1) | Fertiliser Date 2 (julian day) | Fertiliser Amount 2 (kgNha-1) | Fertiliser Date 3 (julian day) | Fertiliser Amount 3 (kgNha-1) | Harvest (julian day) |
|---|---|---|---|---|---|---|---|---|---|---|---|---|---|
| SANDS | south | SB | 91 | 94 | 150 | 2 | 96 | 75 | 126 | 75 | - | - | 238 |
| | | WB | 285 | 288 | 200 | 2 | 57 | 40 | 88 | 160 | - | - | 211 |
| | | WW | 285 | 288 | 220 | 3 | 74 | 40 | 102 | 90 | 130 | 90 | 250 |
| | | WOSR | 212 | 215 | 210 | 2 | 58 | 105 | 86 | 105 | - | - | 210 |
| | north | SB | 75 | 78 | 150 | 2 | 80 | 75 | 110 | 75 | - | - | 258 |
| | | WB | 285 | 288 | 200 | 2 | 57 | 40 | 88 | 160 | - | - | 211 |
| | | WW | 285 | 288 | 220 | 3 | 84 | 40 | 112 | 90 | 140 | 90 | 270 |
| | | WOSR | 212 | 215 | 210 | 2 | 58 | 105 | 86 | 105 | - | - | 210 |
| OTHER | south | SB | 91 | 94 | 130 | 2 | 96 | 65 | 126 | 65 | - | - | 238 |
| | | WB | 285 | 288 | 180 | 2 | 57 | 40 | 88 | 140 | - | - | 211 |
| | | WW | 285 | 288 | 200 | 3 | 74 | 40 | 102 | 80 | 130 | 80 | 250 |
| | | WOSR | 212 | 215 | 210 | 2 | 58 | 105 | 86 | 105 | - | - | 210 |
| | north | SB | 75 | 78 | 130 | 2 | 80 | 65 | 110 | 65 | - | - | 258 |
| | | WB | 285 | 288 | 180 | 2 | 57 | 40 | 88 | 140 | - | - | 211 |
| | | WW | 285 | 288 | 200 | 3 | 84 | 40 | 112 | 80 | 140 | 80 | 270 |
| | | WOSR | 212 | 215 | 210 | 2 | 58 | 105 | 86 | 105 | - | - | 210 |
| SALO | south | SB | 91 | 94 | 130 | 2 | 96 | 65 | 126 | 65 | - | - | 238 |
| | | WB | 285 | 288 | 180 | 2 | 57 | 40 | 88 | 140 | - | - | 211 |
| | | WW | 285 | 288 | 200 | 3 | 74 | 40 | 102 | 80 | 130 | 80 | 250 |
| | | WOSR | 212 | 215 | 210 | 2 | 58 | 105 | 86 | 105 | - | - | 210 |
| | north | SB | 75 | 78 | 130 | 2 | 80 | 65 | 110 | 65 | - | - | 258 |
| | | WB | 285 | 288 | 180 | 2 | 57 | 40 | 88 | 140 | - | - | 211 |
| | | WW | 285 | 288 | 200 | 3 | 84 | 40 | 112 | 80 | 140 | 80 | 270 |
| | | WOSR | 212 | 215 | 210 | 2 | 58 | 105 | 86 | 105 | - | - | 210 |
| SHALLOW | south | SB | 91 | 94 | 150 | 2 | 96 | 75 | 126 | 75 | - | - | 238 |
| | | WB | 285 | 288 | 200 | 2 | 57 | 40 | 88 | 160 | - | - | 211 |
| | | WW | 285 | 288 | 220 | 3 | 74 | 40 | 102 | 90 | 130 | 90 | 250 |
| | | WOSR | 212 | 215 | 210 | 2 | 58 | 105 | 86 | 105 | - | - | 210 |
| | north | SB | 75 | 78 | 150 | 2 | 80 | 75 | 110 | 75 | - | - | 258 |
| | | WB | 285 | 288 | 200 | 2 | 57 | 40 | 88 | 160 | - | - | 211 |
| | | WW | 285 | 288 | 220 | 3 | 84 | 40 | 112 | 90 | 140 | 90 | 270 |
| | | WOSR | 212 | 215 | 210 | 2 | 58 | 105 | 86 | 105 | - | - | 210 |
| HUMOSE | south | SB | 91 | 94 | 80 | 2 | 96 | 40 | 126 | 40 | - | - | 238 |
| | | WB | 285 | 288 | 120 | 2 | 87 | 40 | 88 | 80 | - | - | 211 |
| | | WW | 285 | 288 | 140 | 3 | 74 | 40 | 102 | 50 | 130 | 50 | 250 |
| | | WOSR | 212 | 215 | 130 | 2 | 58 | 65 | 86 | 65 | - | - | 210 |
| | north | SB | 75 | 78 | 80 | 2 | 80 | 40 | 110 | 40 | - | - | 258 |
| | | WB | 285 | 288 | 120 | 2 | 87 | 40 | 88 | 80 | - | - | 211 |
| | | WW | 285 | 288 | 140 | 3 | 84 | 40 | 112 | 50 | 140 | 50 | 270 |
| | | WOSR | 212 | 215 | 130 | 2 | 58 | 65 | 86 | 65 | - | - | 210 |

SB: Spring Barley  WB: Winter Barley  WW: Winter Wheat  WOSR: Winter Oilseed Rape
South - North border is set at 56.64N
SANDS soil type includes soils which are sand, loamy sand or sandy loam to 40 cm depth and are sand or loamy sand between 40 and 80 cm
SALO soil type includes sandy loam and loamy soils
SHALLOW soil type includes soils over impermeable subsoils and those where the parent rock (chalk, limestone or other rock) is within 40 cm of the soil surface
HUMOSE soil type includes soil with 10 and 20% organic matter
OTHER soil type includes all soils that do not fall in any of the other soil types



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
