# Peer review of "Estimating the soil $N_2O$ emission intensity of croplands in northwest Europe"

_Biogeosciences, 2018_

## Referee Comment (RC1) · Anonymous Referee #1 · 6 Feb 2019

The manuscript "Estimating the soil N2O emission intensity of croplands in northwest Europe" describes the estimation of N2O emission factors, relative leaching losses and relative uptake of applied N fertilisation by using the process based model LandscapeDNDC. The results are accompanied by the quantification of modelling uncertainty taking uncertainties of soil properties and model parameters into account. The article is clearly written, e.g. concepts and applied methods are understandable and conclusion sound. The comprehensive comparison of upscaling results with other study is noteworthy as it allows assessing the plausibility and issues of the derived factors. Shortcomings and constraints of the model application are named and critically discussed and might serve ongoing research. Therefore the work is recommended for publication. However some aspects could be addressed: One important aspect of

the novel approach is that exchange via crop uptake and leaching is also considered. The reasonability of leaching losses and uptake are discussed. Discussed N fluxes do not include gaseous losses of NO and N2 and the storage change of NO3, NH4 and organic N in the soil (which seems to be an important element of the N budget). It would have been interesting to see these fluxes and changes too. Those might be useful references for upcoming studies. It was an aim of this study to quantify crop specific N2O emission factors in Scotland. Emission factors modelled here represent one specific year (2013). It is unclear how the climate conditions of this year compare to mean average climate conditions of Scotland. Extraordinary dry or wet conditions might bias the estimated emission factor (also for NO3 and N uptake fractions) and thus restrict its applicability. The uncertainty ranges of parameters described in table 1 are comparable to the uncertainty ranges of the Posterior distributions in Myrgiotis et al. (2018b). In this publication (Myrgiotis et al. (2018b) the effect of parameter distributions on measured N2O emissions is discussed but not the effect on annually aggregated emissions. From figure 4 in Myrgiotis et al. (2018b) it seems that if modelled N2O range failed to cover measured values these measured values were often emission peaks. Therefore my question is: Are model uncertainties of annually aggregated emissions well represented by Monte Carlo analyses using the described parameter ranges. It is also not clear to me what kind of uncertainty range is produced (95% confidence limits, standard deviation). In this chapter correlations between model parameters and between soil properties that have been sampled are not mentioned. For instance, the relationship between bulk density and SOC or SOC and clay content is well known. Independent sampling (not considering these relationships) might affect modelled uncertainty ranges. If parameter ranges are based on Myrgiotis et al. 2018b sampling from the multivariate posterior parameter distribution would consider the correlation between parameters. It might be useful to address these issues in the discussion or describe the methods more clearly in the MM part. It is not mentioned from what kind of distributions sampling was performed (uniform, normal distributions or posterior parameter distributions).
P16/L7: Fitton et al., 2017 not in brackets P16/L16: "dataset compiled by Stehfest and Bouwman" P18/L12: "The low sensitivity of the predicted crop N uptake to model parameters and soil inputs reflects the fact that, under sufficient N supply, climate is the main determinant of crop growth and, consequently, N uptake." , supply instead of suppply, for the importance of soil properties versus climate see Hoffmann et al. 2016

Hoffmann, H., Zhao, G., Asseng, S., Bindi, M., Biernath, C., Constantin, J., ... & Gaiser, T. (2016). Impact of spatial soil and climate input data aggregation on regional yield simulations. PloS one, 11(4), e0151782.

---

## Referee Comment (RC2) · Anonymous Referee #2 · 20 Feb 2019

The manuscript "Estimating the soil N2O emission intensity of croplands in northwest Europe" by Vasileios Myrgiotis et al. evaluates N2O emissions factors as well as N uptake and N leaching factors with the focus on four arable crops in Eastern Scotland with the use of the process-based model Landscape-DNDC. An uncertainty analysis of model predictions associated to soil input and model parameter values used in the simulations is presented too. From a scientific perspective, the manuscript is found to be a sound study that contributes to approach better options for climate change mitigation from regional agriculture production. The methodology, based on modelling, available regional databases (soil, crops, climate and management) and previous studies from the same group and the literature is appropriate to the aims of the study. Conclusions are consistent with the results. The manuscript is well written and results are overall

well presented. The structure of the paper may, however, be improved. For consistency purposes with section 2 (presentation of materials and methods), I suggest that the results start with the section on "Relative uncertainty of model predictions" (currently last subsection in the results). This should be also ok from the storyline perspective. I have also found parts of discussion in the results section (last paragraphs pages 12-13, from line 14, page 14 to line 5 page 15). Also, some sentences in materials and methods are read as an introduction (first sentence in sections 2.3 and 2.4.2). The format of the references needs a general revision (format, missing titles of articles,. . .).

Specific list of issues to be addressed in a revision of the manuscript:

Page 1. Abstract. "Reducing the nitrogen (N) footprint of agriculture is a global challenge that depends on our ability to quantify the N2O emission. . .". But it also depends on many other things. How, more specifically, could the ability to quantify N20 emissions contribute to reduce N footprint?

Page 3. "Especially". I think this word does not add anything here. Suggestion "Model-based estimates of soil N2O depend. . ."

Page 3. Line 14. It is not clear in the sentence why the reference of Haas et al., 2012 is used.

Page 4. Line 5, which specific model version is the "state-of-the-art" version?.

Page 4. Line 7. Is simulated crop yield the grain yield?

Page 4. Line 11, "ammonium nitrate was considered. . .". NH3 volatilization could be a potentially significant flux on the contrary of what is previously reported (page 3, lines 30-31). This should be addressed in in the text.

Page 4. Line 12. WOSR, winter oilseed rape, should be used here first, not in page 10

Page 5. Line 2. "input data on climatic. . ." Time resolution daily, weekly, monthly. . .??

Page 5. Line 11 "their values. . ." Where they come from? Are default values?

Page 5. Line 16. Size of the grid cells?.

Page 10. Line 10. Fig. 4 does not show EFs according to soil texture.

Page 10. Lines 15-16. According to the results the total mean EFs (N2O + NO3 + NU) estimated from N fertilizer would be about 50%. Half of N from fertilizer applications is somewhere else (changes in soil N storage, volatilization (NH3, N2, nitric oxide...), not harvested parts of the crop....This should be considered when discussing the results and conclusions.

Page 12. Lines 5-6. "N loss via leaching and crop uptake act in opposition to the emission of N2O...". The explanation can not be derived from the results in the study.

Page 16. Line 8. According to correlations in Fig. 9, C content and water retention are more strongly correlated with N2O emissions than fertilizer rates. If the authors refer to management factors, please specify.

Page 17. Line 17. How the tendency to underestimate crop N uptake is supported? No direct or indirect evaluation of the simulated WOSR N uptake is done in the study. On the other hand simulated yields of WOSR are very close to measurements (Fig. 4).

Page 17. Line 28 "discrepancy between modelled leaching losses and observations". It is not clear which are the observations that the authors do refer.

Page 18. Line 5. "fertiliser"

Table 1. Please add the pertinent units to the parameters reported.

Figure 4. Change "tn ha-1" by "t ha-1"

---

## Author Comment (AC1) · 20 Feb 2019

Thank you very much for taking the time to review our article and for your comments/suggestions. Below, we respond to the main points of your review.

The first comment refers to the fact that potentially important elements of an agroecosystem's N budget (e.g. NO, N2, soil NO3/NO4/organic-N) are not covered in this study. It is true that presenting model outputs for all the N-based outflows from the plant-soil-atmosphere-water system would have made the study even more useful. We decided to consider the minimum number of N-based system variables that could provide a good (yet not complete) picture of the N budget of arable ecosystems in Eastern Scotland. The three main aspects behind our decision to consider only plant N and

NO3 leaching, in addition to N2O emissions, were : (1) the lack of field measured datasets or sufficient locally-relevant literature on any other N-based variable (e.g. N2) meant that we would have nothing to compare our outputs with (2) any additional model output considered would have increased the size of the raw regional-scale model outputs that need to be processed for final analysis/presentation —that would have had a considerable computational cost (3) discussing and presenting any additional N-based gases or soil elements would have required more text space What weighted more was the fact that both NO and N2 are very difficult to measure on the field [1,2], that measured data for NO/N2 are generally rare and, mainly, that no relevant dataset was available to us. Similarly, and while we had some field measured data for soil NO3/NO4 from UK sites (i.e. sites used in Myrgiotis et al 2018b), we believe that presenting soil NO3/NO4 model outputs would have complicated things significantly. This is mainly (but not exclusively) due to the fact that the Landscape-DNDC model calculates and outputs soil NO3/NO4 on a soil layer-by-layer format, which, in turn, means that for just a single model run at a single point/location multiple values would have had to be read andprocessed —and this would have had a prohibitive computational cost.

The second comment refers to how representative of Eastern Scotland's climate was the weather in the area during 2013. Mean annual temperature across the region was not different from the 1981-2010 mean but 2013 was quite dry when compared to the 1981-2010 mean (https://www.metoffice.gov.uk/climate/uk/summaries/2013/annual - for climate anomaly maps). However, the total precipitation during spring was similar to the 30-years mean. One could argue that drier conditions might lead to emission factors that are lower than what is "normal" for this area. Nevertheless, as we show in Figure 9, no single factor/variable affects soil N2O emissions so strongly that such a statement could be made without hesitation. In conclusion, we believe that adding some text on the climatic representativeness of 2013 would be useful and will be done in the final revision.

The third comment refers to the 9 model parameters that were identified as being crucial for N2O prediction in previous publications (Myrgiotis et al, 2018a,b) and were used in this study. Table 1 presents, for each parameter, the value used to perform the regional simulations and the "realistic" margins that were used in the uncertainty analysis. Thank you for noticing our omission to state that the samples were drawn from the posterior distributions (presented in Myrgiotis et al, 2018b) —we will add a statement about this at the materials and methods section in the final revision. Regarding the uncertainty range that is produced for each output variable (N2O, N uptake, NO3); the uncertainty analysis runs produced 300 values (per output variable and per point/location). We considered different ways to visualise the results of the pseudo-regional uncertainty analysis and concluded that using the relative standard deviation is the best way to present the results for all 3 output variables in a single figure in which readers can see/understand how "sensitive" the regional estimates were to soil inputs and model parameters. Essentially, at each point/location, which was randomly selected and used in the analysis, we divided the standard deviation by the mean of the 300 values produced.

Finally, another issue that was raised regarded the validity/appropriateness of the parameter ranges used in quantifying model output uncertainty. Indeed, according to the results presented in Myrgiotis et al, 2018b, under certain conditions the model might fail to estimate N2O emission peaks when measurements show them occuring. In regards to this, we argue that (1) some of these missed emission peaks were single points with considerable uncertainty attached (i.e. large variability across measured N2O samples) and (2) we identified those "missed-peak" instances as caused by Landscape-DNDC's very detailed soil discretisation method according to which N2O produced at a certain soil layer travels through the soil profile before being released to the atmosphere and can be transformed to other N-gases during this travel; we would like to note that having no or very low emission peaks after fertiliser application is a possibility according to field studies and especially under protracted wet soil conditons. We cannot exclude the possibility that in some of the thousands of point-runs performed, this no/low emission peak phenomenon has occurred. However, plots of time series of the

mean (all grid cells) daily N2O emissions (per crop type) showed the peaks/troughs that one generally expects before/after fertiser application, which means that any no/low peak instances, if present in the our results, were very rare (e.g. no/low peak instances cause mean daily N2O to become an elevated flat line with no peaks/troughs). Overall, we believe that Myrgiotis et al, 2018b provided a field-data-based "constraining" of the theoretical/default parameter ranges and that this constraining was as good as the data themselves (and bayesian calibration and model structure) allowed.

1. Menidets et al, 2015 - A review of soil NO transformation: Associated processes and possible physiological significance on organisms 2. Butterbach-Bahl et al, 2013 - Nitrous oxide emissions from soils: how well do we understand the processes and their controls?

---

## Author Comment (AC2) · 28 Feb 2019

The authors would like to thank the second anonymous reviewer for his comments and recommendations. Below we respond to each of his/her comment (P : page - L : line).

On the structure of the article :
(1) We believe that we should keep the current structure of the article; instead of first presenting uncertainty in the results section . This is because our current introduction (P3 last paragraph and points 1-3 in P3:4) lets the reader know that in our results and discussion we will, first, address N2O EFs, then, NO3 and N uptake and finally regional-scale uncertainty.
(2) Regarding some sentences in our results sub-sections that read as being part of the

**BGD**

introduction. For the first sentence in 2.4.2; the sentence was added in response to a comment by the editor and there is nowhere else in the document that this information can be added without affecting the text's flow. For the first sentence in 2.3; we followed the reviewer's recommendation and removed the sentence in the revised document.

(3) Regarding parts of the discussion that the reviewer found in the results section, we believe that these are very brief discussions to complement the presentation of results. Results are, then, discussed in detail in the discussion section.

On the errors in the references:
Indeed, there were a number of errors in the reference list which we corrected in the revised document.

On the specific issues raised (see middle of C2 and onwards, in discussion paper):
> Abstract : The ability to quantify the N2O emission intensity of croplands and crop management is one of the things that can help us reduce the N footprint. We added "among other things" to the abstract. However, there is not enough space, in the abstract, to discuss how the quantification can be used to reduce the N footprint. We believe this is discussed extensively in our discussion and conclusions.
> P3 : Removed "especially"
> P3 - L14 : Removed reference to "Haas et al, 2012"
> P4 - L5 : The model, overall, represents the state-of-the-art in agro-ecosystem biogeochemistry modelling. We believe that there is no reason do discuss specific model versions because it is difficult and text-demanding to discuss/present the differences/advances between different versions.
> P4 - L7 : Yes, the simulated yield is the amount of grain removed from the field so that it can be compared with Scotland's agricultural census statistics.
> P4 - L11 : This relates to NH3 volatilisation; an issue that was raised by the editor and reviewer 1. We revised the text in which we refer to this issue (last paragraph in P3). Nitrogen losses via NH3 volatilisation are, no doubt, important but dealing

with this N-based outflow was beyond the scope of our study. We also believe that well-managed, rainfed, NW European croplands that are treated only with ammonium nitrate are the fields least prone to NH3-volatilisation (Pan, B. et al. 2016. Ammonia volatilization from synthetic fertilizers and its mitigation strategies: A global synthesis, Agriculture, Ecosystems and Environment)

> P4 - L12 : Winter oilseed rape abbreviated here (WOSR) as suggested.

> P5 - L2 : Information added (i.e. daily).

> P5 - L11 : We added text referring to the fact that parameter ranges and values used came from a previous study where the model was calibrated (Myrgiotis et al 2018b).

> P5 - L16 : Information added (1km-2)

> P10 - L10 : Fig 4 now shows EFs according to soil texture.

> P10 - L15:16 : We added text and citations (P18 - L5:8) discussing/referring to this issue in the revised document.

> P12 - L5:6 : The transformation/translocation of N via leaching, gasification and uptake by crops are processes that act against each other because if a mole of N present in the soil is not leached it might be gasified, emitted to the atmosphere or absorbed by the crop. The order in which these processes can happen in a cubic millimeter of soil is not fixed since they are competing processes and depend on numerous factors, which act at minuscule scale. In the modelled system however, this competition is non existent. Whichever process (gasification, leaching, crop uptake) is modelled/simulated first has an "advantage" over the remaining processes. What we mean by using the sentence in question is that the model estimates N uptake and leachate formation before gas formation. We refer to this in the revised document (P13 - L5:6).

> P16 - L8 : We do not refer to management factors

> P17 - L5:18 : The tendency of the model to underestimate crop N uptake by WOSR is supported mainly by the fact that WOSR is not a cereal but (in essence) it is simulated as being one, in Landscape-DNDC. WOSR expects, on average, half of its total N need to be covered right after the 1st fertiliser application which reflects its

"rapid" (or faster-than-a-cereal) post-winter N-demand pattern. On the other hand, a winter cereal expects roughly 20% of its total N needs to be covered right after the 1st fertiliser application; and the remaining 80% to be covered after the one or two subsequent fertiliser applications (see appendix table). Therefore, it is possible that the simulated WOSR (because it grows more like a cereal) cannot take up all the N available in the soil (after the 1st fertiliser application). WOSR will eventually take up enough N, later on, to reach a yield level similar to that observed in the statistics. On this same issue, we argue that WOSR is a N-hungry crop that does not produce as much yield-biomass as cereals do. We believe that the official fertiliser use guide suggests the application of "rather" ample quantities of N to fields under WOSR. We assume that these recommendations are based on field experiments.

> P18 - L5 : typo corrected

> Table 1 : units corrected

> Figure 4 : typo corrected

Please also note the supplement to this comment:
https://www.biogeosciences-discuss.net/bg-2018-490/bg-2018-490-AC2-supplement.pdf

―――――――――――――――――――――――――

**Supplement:**

[revised manuscript text omitted]

---

## Author Response (AR1)

Dear Editor,

Thank you for the prompt review of our response to the referees' comments.

We have made changes in response to the request for clearly separating introduction, methods, results and discussion. Based on the comments of Referee #2, and considering the changes we have already made, we understand that the main remaining issue was that we briefly discuss our results in the results section. This happens in the "Correlation of model drivers and outputs" and the "Relative uncertainty of model predictions" sub-sections. In the newly revised document we have moved any discussion text from these two results sub-sections to the document's discussion section. See P12 - L5 to P13 - L13 for where text was removed and P17 - L24 to P18 - L17 for where text has been added/adjusted. We hope these revisions are satisfactory but remain open to further editing the text if that is considered necessary.

Kind regards,
The authors